# The Impact of Nutrition Education, Strength Training, and Body Image Perception on Orthorexia Nervosa Risk: A Cross-Sectional Study in Women

**DOI:** 10.3390/bs15020199

**Published:** 2025-02-13

**Authors:** Wiktoria Staśkiewicz-Bartecka, Laura Tambor, Agata Kiciak, Daria Dobkowska-Szefer, Natalia Kuczka, Agnieszka Białek-Dratwa, Agnieszka Bielaszka, Oskar Kowalski, Marek Kardas

**Affiliations:** 1Department of Food Technology and Quality Evaluation, Department of Dietetics, Faculty of Public Health in Bytom, Medical University of Silesia in Katowice, 41-808 Zabrze, Poland; ltambor@sum.edu.pl (L.T.); akiciak@sum.edu.pl (A.K.); ddobkowska-szefer@sum.edu.pl (D.D.-S.); natalia.k4940@gmail.com (N.K.); abielaszka@sum.edu.pl (A.B.); mkardas@sum.edu.pl (M.K.); 2Department Human Nutrition, Department of Dietetics, Faculty of Public Health in Bytom, Medical University of Silesia in Katowice, 41-808 Zabrze, Poland; abialek@sum.edu.pl (A.B.-D.); okowalski@sum.edu.pl (O.K.)

**Keywords:** orthorexia nervosa, body image perception, body self-esteem, dietetics students, eating disorders

## Abstract

Background: The increasing focus on healthy eating and achieving an ideal physique can lead to the development of disordered eating patterns, including orthorexia nervosa. The relationship between body image perception, self-esteem, and the risk of orthorexia nervosa is multifaceted, with negative body image and low self-esteem often acting as contributing factors. This study aimed to assess the risk of orthorexia nervosa and analyze body image perception among women engaged in strength training and dietetics students. Methods: The study was conducted using a Computer-Assisted Web Interview with 190 women aged 18 and older, divided into four groups: dietetics students engaged in strength training, dietetics students not engaged in strength training, non-dietetics women engaged in strength training, and non-dietetics women not engaged in strength training. The study utilized the Düsseldorf Orthorexia Scale to assess the risk of orthorexia nervosa and the Body-Esteem Scale to evaluate body image perception. Results: The highest orthorexia nervosa risk was observed in dietetics students who do not engage in a strength training group (60.9%), followed by dietetics students who engage in a strength training group (37.5%). In contrast, women who are not studying dietetics and do not engage in a strength training group exhibited the lowest orthorexia nervosa risk (13%), emphasizing the protective role of physical activity and reduced focus on nutritional rigidity. Regression analysis revealed that low appearance self-esteem (*p* = 0.011) and high social attribution (*p* = 0.043) significantly predicted higher orthorexia nervosa risk. Group affiliation also influenced orthorexia nervosa risk, with dietetics students showing higher Düsseldorf Orthorexia Scale scores. However, weight-related self-esteem (*p* = 0.082) did not significantly impact orthorexia nervosa scores. Conclusions: Dietetics education and physical activity independently and collectively affect orthorexia nervosa risk. Students in dietetics, particularly those not engaged in strength training, are at a higher risk due to the dual influence of academic pressures and heightened nutritional awareness. The findings underscore the need for interventions targeting body image perception, promoting flexible dietary approaches, and addressing external validation pressures to mitigate orthorexia nervosa risk in high-risk groups.

## 1. Introduction

### 1.1. Background and Definition of Orthorexia Nervosa

Lifestyle significantly impacts health, contributing up to 50% of one’s health status, as highlighted by Lalonde. A healthy lifestyle encompasses proper nutrition, regular physical activity, adequate sleep, stress management, and avoiding harmful substances ([20]). However, the increasing emphasis on healthy living and idealized body standards has also led to a rise in eating disorders (EDs), affecting 5–18% of the population, particularly among young individuals. Orthorexia nervosa (ON) is one such disordered eating behavior, estimated to affect 7% of the population ([30]). It is an obsession with healthy eating, where the individual places extreme importance on the quality of the food consumed. The disorder also manifests as a tremendous fear of eating anything considered unhealthy. There are many causes of eating disorders, but a common one is a distorted perception of one’s body, which may be related to the idealized body images presented in social media ([19]; [26]).

### 1.2. Diagnostic Classification and Challenges

Although the topic of ON is increasingly discussed in the media, it is still not recognized as a separate disorder in the current International Statistical Classification of Diseases and Related Health Problems (ICD-11), which is in force in Europe. The classification used in the United States—DSM-V (Diagnostic and Statistical Manual of Mental Disorders, 5th Edition)—also does not include this phenomenon ([15]; [21]). The lack of classification may result from the multidimensionality of ON, which includes elements characteristic of both EDs and obsessive compulsive disorders (OCDs). On the one hand, ON resembles an ED, such as anorexia nervosa (AN), through obsessive thoughts related to food. On the other hand, its obsessive compulsive aspects, e.g., rituals surrounding food preparation or avoiding “unhealthy” products, may indicate a connection to OCD ([9]; [13]). This creates difficulties in definitively assigning ON to a specific diagnostic category and raises questions about whether it should be considered a subtype of ED, a subtype of OCD, or a completely separate diagnostic entity ([15]; [21]). An important milestone in the field was marked by the 2022 publication by Donini LM et al., which involved an international collaboration to develop a more standardized framework for diagnosing ON ([8]). Despite its significance, the proposed criteria were derived from expert opinions rather than robust empirical data, highlighting a key limitation. Moreover, the study’s scope excluded numerous global regions, restricting its applicability across diverse populations. Nevertheless, the article represented a major advancement by offering initial, broadly accepted diagnostic guidelines for ON, laying the groundwork for further studies and facilitating progress in the recognition and management of this condition. Recent advancements in the literature highlight important nuances in understanding the overlap between AN and OCD. [28] ([28]) underscore that this relationship is mediated by food-related concerns, which play a pivotal role in the cognitive and emotional processes linking these conditions. This insight provides an important framework for considering targeted interventions that address these mediating factors to improve outcomes for individuals affected by both disorders ([28]).

### 1.3. Risk Factors, At-Risk Groups, and Body Image Concerns

ON is also observed among individuals with high levels of physical activity, as well as those striving to achieve the ideal physique promoted in the media. A slim and muscular figure is often considered a symbol of the ideal lifestyle, with extremely healthy eating as the process leading to that goal. Research available in the current scientific literature suggests the existence of particularly at-risk groups for the development of ON. These groups include individuals whose professions are closely linked to maintaining a healthy body and physique. These include athletes, dietitians, healthcare workers, medical students, trainers, and individuals who have previously suffered from EDs ([4]; [19]). The study by Brytek-Matera et al. examined the relationship between ON traits and body image, fitness, and health attitudes in university students with normal body weight. Among female students with ON, higher body satisfaction and lower concerns about fitness and appearance predicted greater fixation on healthy eating, while no significant associations were found for male students ([4]). Similarly, Dell’Osso et al. reported a higher prevalence of ON symptoms among women, with over one-third of Italian university students showing signs of orthorexia ([7]). Malmborg et al. found that ON prevalence was highest among exercise science students, particularly men, and lowest in business students, with physically active women and dietetics students identified as high-risk groups ([23]). Body image, a multidimensional concept encompassing how individuals perceive, feel, and behave toward their bodies, plays a significant role in personal identity. Distorted body image perception is widespread and often leads to harmful behaviors, including unhealthy diets, low self-esteem, poor mood, and difficulties in social or professional functioning, which can escalate into severe disorders such as body dysmorphia, anorexia nervosa, or bulimia nervosa ([23]; [32]). Social media significantly influences modern beauty standards, offering platforms for education and community building but also promoting idealized and often unrealistic body images ([5]). This can lead to body dissatisfaction, mental health issues, and unhealthy eating behaviors, with platforms like Instagram linked to lower self-esteem and a greater concern over body image ([10]; [39]). Research confirms that the use of applications such as Instagram is associated with significant concerns about body image, potentially resulting in lower self-esteem. Regular browsing of images and videos is likely crucial to one’s body image ([18]; [29]). Women often engage in physical activity primarily to achieve goals like weight loss or body shaping rather than for enjoyment, with gym workouts popular for improving appearance and self-esteem ([14]; [34]).

### 1.4. Psychodynamic Perspectives on Orthorexia Nervosa

Beyond external pressures, psychodynamic theories suggest that ON may serve as a coping mechanism for deeper psychological conflicts ([17]). The pursuit of dietary purity often reflects an internalized belief linking self-worth to food choices, as individuals use rigid dietary behaviors to exert control and address feelings of inadequacy or instability. Understanding these emotional underpinnings is crucial for effective prevention and intervention strategies ([11]; [12]; [25]).

### 1.5. Research Gap, Objectives, and Hypothesis

This study aims to address a significant research gap stemming from inconclusive findings regarding the relationship between a health-oriented lifestyle and the risk of developing ON in women. Although previous research ([4]; [7]; [23]) has indicated correlations between the pursuit of the ideal body shape and unhealthy eating patterns, the roles of specific factors—such as participation in strength training and education in dietetics—remain insufficiently clarified.

Accordingly, a research question was formulated: Does diet education and regular strength training increase the risk of developing ON in women, and does self-perception of the body play a mediating role in this relationship? To answer this question, the objectives of the study included assessing the risk of ON and analyzing attitudes toward body image among women, including participants in strength training and female dietetics students, assessing the nutritional status of the subjects, comparing the level of ON risk and body perception between groups, and analyzing the relationship between ON risk and multidimensional self-perception.

The research hypothesis is that both diet education and regular strength training contribute to an increased risk of developing ON in women, with a distorted self-image of the body being an important mediating mechanism for this relationship.

## 2. Materials and Methods

### 2.1. Survey Design

The study was conducted using the CAWI method (Computer-Assisted Web Interview) between April and May 2024. The questionnaire was distributed via a link in social media groups related to women, dietetics, and strength training. The study employed a purposive sampling method. To ensure data reliability, completed questionnaires were filtered to eliminate incomplete responses. This helped to exclude potentially unreliable data, increasing the credibility of the study’s results.

### 2.2. Study Participants

The study involved 204 women who were divided into four groups:SDNTS—dietetics students who do not engage in strength training (*N* = 46);SDTS—dietetics students who engage in strength training (*N* = 48);NDNTS—women who are not studying dietetics and do not engage in strength training (*N* = 50);NDTS—women who are not studying dietetics but engage in strength training (*N* = 46).

The inclusion and exclusion criteria for the study are shown in Figure 1.

Strength training was defined as structured workouts focused on resistance exercises, such as weightlifting, bodyweight exercises, or training with resistance bands, performed in gyms, fitness clubs, or at home. After considering the exclusion criteria, 190 participants were included in the final analysis.

All participants in the study were thoroughly informed about its purpose and assured of their anonymity, with their consent obtained for data usage. Details about the voluntary and informed nature of their participation were provided at the start of the survey. This study was conducted in compliance with the Declaration of Helsinki, as set out by the World Medical Association. Ethical approval was granted by the Bioethics Committee of the Medical University of Silesia in Katowice (BNW/NWN/0043-3/641/35/23, approval date: 22 September 2023) in accordance with the Act of 5 December 1996, concerning the professions of physicians and dentists ([27]).

### 2.3. Research Tools

The risk of ON and body image assessment were evaluated using a questionnaire consisting of demographic information and two validated scales: the Body-Esteem Scale for Adolescents and Adults (BESAA) and the Düsseldorf Orthorexia Scale (PL-DOS). The demographic section contained general questions regarding group affiliation, age, body weight, height, education, place of residence, health conditions, and eating behaviors. Additionally, the demographic section included a question on diagnosed chronic conditions, specifically mental health disorders such as depression, eating disorders, and neurosis. None of the respondents indicated having any chronic mental health conditions.

#### 2.3.1. DOS

The DOS is a screening tool designed to assess orthorexic eating behaviors ([2]). In this study, the respondents completed the 10-item DOS, answering on a four-point Likert scale ranging from “definitely not applicable to me” to “definitely applicable to me”, with no reverse-scored items ([3]). The maximum score was 40 points, and the interpretation of the results was as follows: a score over 30 points indicated the presence of ON, scores between 25 and 29 points suggested a risk of ON, and a score below 25 points indicated no presence of ON ([2]). The study utilized the Polish adaptation of the DOS (PL-DOS) ([3]), which demonstrated reliability comparable to the original E-DOS, with a Cronbach’s α coefficient of 0.84 ([2]; [3]).

A reliability analysis using ω McDonald’s was conducted on the results obtained from the PL-DOS scale. The obtained score was 0.874, indicating a high level of internal consistency for the scale.

#### 2.3.2. BESAA

The BESAA scale provides a multifaceted analysis of attitudes toward one’s body, encompassing specific aspects related to weight, general appearance, and social perceptions ([31]). The BESAA scale consists of 23 items divided into three subscales.

The Appearance Self-Esteem Subscale (BE—Appearance) includes 10 items related to general feelings about one’s physical appearance, offering insights into participants’ satisfaction with their overall look. The weight self-esteem subscale (BE—Weight) consists of 8 items focusing on satisfaction with one’s weight. The Attribution Self-Esteem Subscale (BE—Attribution) comprising 5 items, addresses beliefs about how others evaluate one’s appearance.

Respondents provide answers on a five-point Likert scale, ranging from 1 (never) to 5 (always). Higher scores on each subscale indicate more positive attitudes toward a particular aspect of body image. In contrast, lower scores indicate dissatisfaction ([31]).

A reliability analysis using ω McDonald’s was conducted on the results obtained from the BESSA scale. The obtained score was 0.834, indicating good internal consistency for the scale.

#### 2.3.3. BMI (Body Mass Index)

The participants’ nutritional status was assessed using the BMI, calculated by dividing body weight in kilograms by the square of height in meters (BMI = weight (kg)/height (m^2^)). The resulting BMI values were interpreted according to the WHO classification and according to the WHO guidelines ([38]).

#### 2.3.4. Statistical Analysis

Statistical analyses were performed using Statistica software version 13.3 (StatSoft Poland, Kraków, Poland). The values of the measurable variables were presented as arithmetic means (X) with standard deviation (SD), minimum (Min), maximum (Max), and median (Med). For non-measurable variables, frequencies (N) and percentages (%) were used. The normality of the data distribution was assessed using the Shapiro–Wilk test.

A one-way analysis of variance (ANOVA) was applied to evaluate statistically significant differences in continuous variables, such as BMI and body image assessment subscales (BE—Appearance, BE—Weight, BE—Attribution), across the defined study groups. The chi-square test (χ^2^) was used to analyze relationships between categorical variables, including group affiliation and the risk of ON assessed by the DOS scale, as well as BMI classification.

In the statistical analysis, effect sizes were calculated as a complement to *p*-values. For comparisons of continuous variables, ε^2^ was used. For categorical variables, Cramér’s V was applied to assess the strength of associations.

Linear regression analysis was employed to assess the impact of predictors such as BESAA subscale scores and group affiliation on the risk of ON, as measured by the DOS scale. The reliability of the scales used in the study (e.g., BESAA, DOS-PL) was evaluated using McDonald’s ω coefficient to ensure internal consistency. Additionally, standardized coefficients were calculated to allow for direct comparison of the relative influence of each predictor of ON risk.

A *p*-value of less than 0.05 was considered statistically significant.

## 3. Results

One hundred and ninety women participated in the study after taking into account the inclusion criteria. According to the criteria, the women were divided into the following groups:SDNTS—dietetics students not engaged in strength training (*n* = 46);SDTS—dietetics students engaged in strength training (*n* = 48);NDNTS—non-dietetics students not engaged in strength training (*n* = 50);NDTS—non-dietetics students engaged in strength training (*n* = 46).

The primary sources of nutrition knowledge for the respondents were the Internet (86.8%, *N* = 165) and the scientific literature (54.2%, *N* = 103). Respondents also indicated that dietitians (29.5%, *N* = 56), friends (28.9%, *N* = 55), and trainers (23.2%, *N* = 44) were important sources of information. The characteristics of the study group are presented in Table 1.

Of all the study participants, 70% (*N* = 133) had a normal body weight. Individuals who were overweight accounted for 20% (*N* = 38) of the group, while 5.8% (*N* = 11) were classified as underweight. A statistical relationship was demonstrated between BMI classification and the different respondent groups. Non-dietetics students who did not engage in strength training had the highest percentage of non-normative body weight, according to the BMI classification. Detailed information is presented in Figure 2.

### 3.1. Düsseldorf Orthorexia Scale (DOS-PL)

Based on the DOS questionnaire, orthorexic behaviors were observed in 14 (29.2%) of the strength-training dietetics students and 6 (13%) of the non-dietetics students who engaged in strength training. The group at the highest risk of orthorexia comprised non-strength-training dietetics students (*N* = 28, 60.9%), followed by strength-training dietetics students (*N* = 18, 37.5%) and non-dietetics students engaged in strength training (*N* = 13, 28.3%). The group that showed the least risk of orthorexia consisted of non-dietetics students who did not engage in strength training. These patterns suggest that both education and physical activity contribute to differences in dietary attitudes and orthorexic risk, supporting the study’s hypotheses. The details are shown in Table 2.

There were also statistically significant correlations between the number of scores on the DOS questionnaire and the different groups (*p* < 0.001). Individuals in the SDNTS group achieved an average score of 25.2 ± 3.21, while the SDTS group obtained a slightly higher average of 26.3 ± 4.79. Those in the NDNTS group reached a significantly lower average score of 17.9 ± 4.84, and individuals in the NDTS group obtained an average of 24.1 ± 5.40. Statistically significant relationships were demonstrated between the DOS questionnaire results and the individual respondent groups (*p* < 0.001) (Figure 3).

### 3.2. Body Self-Assessment Scale for Adolescents and Adults (BESAA)

The analysis of results revealed statistically significant differences between groups in levels of self-assessment of appearance (BE—Appearance) (*p* < 0.001), weight (BE—Weight) (*p* < 0.001), and attribution (BE—Attribution) (*p* = 0.006). The results suggest that non-dietetics students, especially those not engaged in strength training, have higher self-assessment in terms of appearance, weight, and attribution compared to dietetics students, which may indicate an influence of dietetics studies and engagement in strength training on body perception. Detailed information is presented in Table 3.

### 3.3. Risk of ON and Assessment of One’s Own Body

The regression analysis showed that self-assessment of appearance (BE—Appearance) and attribution (BE—Attribution) significantly impact the DOS score. Higher self-assessment of appearance was associated with a lower DOS score (*p* = 0.011), while higher attribution increased the DOS score (*p* = 0.043). The effect of weight self-assessment (BE—Weight) was not statistically significant (*p* = 0.082).

The regression model demonstrated moderate explanatory power, with R^2^ = 0.480. These results suggest that both individual factors, such as self-assessment of appearance and attribution, as well as group membership, significantly impact the DOS score (Table 4).

The results of the linear regression analysis present the impact of various factors on ON risk, measured by the DOS scale. The model includes predictors related to body image perception (BESAA), dietetics student status, and strength training engagement.

The analysis revealed that appearance self-assessment (BE—Appearance) is a significant predictor of ON risk, with a coefficient estimate of −0.266 (*p* = 0.002). Conversely, attribution self-assessment (BE—Attribution) was found to have a positive correlation with ON risk, with a coefficient of 0.216 (*p* = 0.043). In contrast, weight self-assessment (BE—Weight) did not show a significant impact on ON risk (coefficient = −0.135, *p* = 0.144).

Additionally, dietetics student status was a significant predictor, with a coefficient estimate of 3.183 (*p* < 0.001). Similarly, strength training status was a significant factor, with a coefficient estimate of 2.797 (*p* < 0.001).

The model demonstrated a moderately strong correlation between the variables and ON risk (R = 0.676), explaining approximately 45.7% of the variance in ON scores (R^2^ = 0.457) (Table 5).

Figure 4 illustrates the relationships between DOS scores (ON risk) and the BESAA subscales (BE—Appearance, BE—Weight, BE—Attribution) across the study groups. The analysis shows that higher BE—Appearance and BE—Weight scores, reflecting greater satisfaction with appearance and weight, are associated with lower DOS scores, indicating a reduced ON risk.

## 4. Discussion

With the increasing awareness of a healthy lifestyle, more and more people are striving to maintain proper eating habits and regular physical activity. However, excessive focus on the quality of consumed food and the pressure to achieve an ideal physique can lead to the development of eating disorders, such as ON. This disorder, although not yet officially recognized by major diagnostic classifications, is becoming the subject of growing interest among scientists and healthcare professionals. Studies have shown that women engaged in strength training and dietetics students are particularly vulnerable to the development of eating disorders, including orthorexia, and low body self-esteem due to increased nutritional awareness, which can become obsessive, and growing societal pressure related to appearance ([13]).

### 4.1. Key Findings and Group Comparisons

The results reveal a significant relationship between the risk of ON and different subgroups, with dietetics students showing particularly elevated risks. The highest ON risk was observed among dietetics students who do not engage in strength training (60.9%), followed by those who engage in strength training (37.5%). This trend may stem from the intensive focus on food quality and health standards inherent to dietetics education, which could foster excessive preoccupation with food purity and dietary control. Notably, non-dietetics students who engage in strength training exhibited a substantially lower risk of ON (13%), aligning with prior research indicating that while physical activity promotes overall health, an obsessive emphasis on food quality can heighten vulnerability to disordered eating behaviors ([37]). Additionally, dietetics students not engaged in strength training might face compounded academic pressures, further exacerbating their focus on nutrition as a coping mechanism. These findings underscore the interplay of educational, physical, and psychological factors in shaping orthorexia risk profiles.

### 4.2. Social Media, Societal Influences, and Nutritional Status

Research by Villa et al. demonstrated that intense use of Instagram is associated with a heightened risk of developing eating disorders, including ON, particularly due to the pressure to achieve the ideal physiques promoted on social media platforms ([37]). These findings align with the current study’s observations, which show that dietetics students and women engaged in strength training—groups more likely to encounter such content—exhibit a higher risk of ON. In the present research, the participants’ dietary behaviors and body image perceptions further underscore the influence of societal and social media-driven ideals, supporting the link between these factors and an increased susceptibility to ON.

The association between social media use and ON risk appears to be consistent across various populations, not limited to Poland. For example, research by Turner and Lefevre in the United States identified a correlation between Instagram use and ON risk, particularly among individuals with perfectionistic tendencies and a focus on appearance ([35]). Similarly, Valente et al. highlighted the prevalence of ON-related content on social media platforms such as Instagram and Twitter, suggesting that these platforms may amplify disordered eating behaviors by promoting idealized dietary practices and body images ([36]). These studies reinforce the findings of Villa et al., suggesting that the impact of social media is a universal phenomenon, transcending cultural and regional boundaries. Such findings emphasize the need for targeted interventions, including education on critical media literacy and support for reducing the negative effects of social media on vulnerable populations.

In the present study, the sample consisted of dietetics students who engage in strength training, dietetics students who do not engage in strength training, non-dietetics individuals who engage in strength training, and non-dietetics individuals who do not engage in strength training. As much as 70% of the study group had a normal body weight, 24.1% were overweight, and 5.8% were underweight. These results are consistent with the findings of Matusik et al., who showed that 75% of their study group had a normal body weight, 14% were overweight, and 8% were underweight ([24]).

### 4.3. Self-Esteem, Body Image, and Eating Behaviors

Based on the DOS scale in the present study, 12.6% (*N* = 24) of the respondents scored as having orthorexia, of which 14 were dietetics students who engage in strength training, and 6 were non-dietetics individuals who engage in strength training. The risk of ON was found in 32.6% (*N* = 62) of the entire group. A correlation between the risk of ON and body image assessment was also confirmed ([1]).

Dietetics students, both those who engage in strength training and those who do not, formed a group with above-average nutritional knowledge and a normal BMI, yet they frequently reported concerns, shame, and dissatisfaction related to their weight and body image in the BESAA questionnaire. These respondents also expressed a strong desire to change their bodies, which confirms the link between low self-esteem and the risk of developing ON. The results suggest that low self-esteem drives unhealthy, obsessive behaviors related to diet and physical activity, which can lead to serious eating disorders ([22]).

Similar conclusions were drawn by Sohind et al., who found that women with a normal BMI who strive for the ideal body promoted by the media often experience lowered self-esteem and a decline in quality of life when they are unable to meet these standards ([33]). Such behaviors may lead to the development of ON, as confirmed in the research by Hafstad et al., who found a high prevalence of ON among physically active individuals (55.3%), suggesting that an obsession with healthy eating is particularly visible in the group of people who regularly engage in sports ([16]).

The results of the BESAA analysis reveal significant differences between groups in terms of BE—Appearance, BE—Weight, and BE—Attribution. The highest scores in appearance and weight self-assessment were recorded in the NDNTS group, suggesting a higher level of acceptance of one’s appearance and less pressure associated with achieving an ideal physique. Conversely, the lowest scores in these scales were observed in the SDNTS group. This may indicate that studying dietetics without regular physical activity is associated with a higher susceptibility to critical self-assessment of one’s body, possibly due to intensive exposure to knowledge about healthy eating and its potential influence on body perception.

Cragun et al. conducted a study to examine the psychometric properties of the BESAA scale among early adolescents, focusing on its ability to measure body image and self-esteem. The findings confirmed that the scale’s subscales, particularly those assessing weight and appearance self-esteem, demonstrated strong internal consistency and convergent validity when compared to measures of self-esteem and BMI. These results highlight the utility of the BESAA scale in exploring body image perceptions in populations susceptible to body dissatisfaction. Similarly, the current study utilized the BESAA scale to assess body image in dietetics students and women engaged in strength training, revealing significant associations between body image dimensions and the risk of ON. Consistent with Cragun et al.’s findings, the weight self-esteem subscale in this study was particularly relevant, as dissatisfaction with weight correlated with increased ON risk. These parallels underscore the scale’s reliability and value in assessing body image across diverse populations and contexts ([6]).

The study results also suggest that non-dietetics groups, especially those not engaged in strength training, have higher self-assessment scores across all subscales compared to dietetics students. These differences may stem from the fact that dietetics students are more aware of health and aesthetic standards concerning the body, which may lead to increased self-criticism regarding their appearance and elevated stress levels associated with not achieving the “ideal” physique.

Regression analysis indicates that BE—Appearance and BE—Attribution significantly influence the DOS score. Higher appearance self-assessment correlates with a lower DOS score, indicating that women who have a more positive view of their appearance are less susceptible to developing ON. Conversely, a higher BE—Attribution score is associated with an increased DOS score, which may mean that individuals who are more sensitive to how they are perceived by others may be more prone to the risk of ON. These findings align with previous research, which suggests that low self-esteem and excessive focus on social perception may increase the risk of EDs ([1]; [16]).

Interestingly, BE—Weight self-assessment did not have a significant impact on DOS scores. This may indicate that body weight-related self-esteem is not as strong a predictor of ON risk as the general appearance self-assessment and social attribution.

Another significant finding is that dietetics student status and engagement in strength training are significant predictors of ON risk. Dietetics students and individuals engaged in regular strength training have higher DOS scores. This may mean that both groups are more susceptible to unhealthy behaviors related to eating, possibly because these individuals are more aware of healthy eating principles and often focused on achieving an ideal physique, which may lead to excessive control over the quality of consumed food and the development of ON.

It is also worth noting the moderate explanatory power of the regression model (R^2^ = 0.480). Although the model shows a moderate correlation between the variables and DOS scores, additional factors that may influence ON risk, such as the influence of social media, social support, and stress levels, should also be considered.

### 4.4. Theoretical Implications and Underlying Mechanisms

Dietetics education and strength training can influence the risk of ON, as both fields emphasize a healthy lifestyle and body shaping, which can foster strict attitudes toward food. Dietetics students typically gain advanced knowledge about nutritional values, dietary components, and the potential harms of unhealthy eating choices. This awareness may lead to an excessive focus on the “purity” of food and the avoidance of items considered “unhealthy”, which is characteristic of ON. In the context of strength training, the additional emphasis on appearance and physical achievements can intensify the drive for control over diet and body, especially in individuals with specific physique goals.

However, the impact varies individually, as each person has unique psychological predispositions and life experiences that may either increase or reduce susceptibility to ON. For example, individuals with high levels of perfectionism or low self-esteem may respond more strongly to strict dietary rules, as diet control provides a sense of value and stability. Conversely, those with strong coping skills may be less likely to develop ON, despite intensive dietetics education or training. Differences in motivation are also essential to consider; while some people may adhere to a strict diet due to external pressures, others may do so based on internal beliefs or personal values.

### 4.5. Practical Implications and Recommendations

The findings of this study highlight the need for practical recommendations to address the risk of ON among dietetics students and women involved in strength training. Specific strategies could include promoting flexible dietary approaches, where participants learn to embrace a balanced variety of foods without excessive restrictions. Incorporating mindful eating practices within dietetics education might help mitigate obsessive thoughts around food purity, reducing guilt and enhancing satisfaction with eating experiences. Practical meal planning exercises that emphasize both health benefits and enjoyment could foster a balanced view of nutrition, encouraging realistic dietary choices. Additionally, providing psychological support focused on improving self-esteem and managing perfectionistic tendencies may be beneficial, as these factors have been shown to contribute to ON development.

Cultural context also plays a significant role in orthorexia tendencies, as societal norms around health and body image vary. In Western cultures, where slim and “clean” diets are highly valued, there may be stronger pressure to maintain restrictive eating habits to achieve an ideal physique, increasing ON risk. This study’s findings indicate that dietetics students, particularly those not involved in strength training, experience a heightened focus on dietary purity, potentially influenced by cultural expectations of health and appearance in their field. In contrast, cultures with more diverse body ideals and less emphasis on diet purity may present a lower risk of ON. Understanding these cultural differences is essential for tailoring interventions that address the specific social and psychological contexts that contribute to ON, ensuring they are relevant and effective across various demographic groups

### 4.6. Strengths and Limitations of the Study

The strengths of the study primarily stem from the carefully selected study groups, which allowed for the analysis of the impact of both strength training and dietetics education on the risk of orthorexia and body image perception. The use of validated research tools, such as the DOS scale to assess the risk of orthorexia and the BESSA scale to assess body image, increases the credibility of the results and allows for comparisons with other studies. The inclusion of multiple variables, such as the assessment of ON risk, body image perception, and nutritional status, provides a more comprehensive view of the issue of EDs.

The study also has certain limitations, primarily stemming from its reliance on self-reported data. One significant concern is social desirability bias, where participants may modify their responses to align with societal norms or expectations. Given the sensitive nature of topics such as eating behaviors and body image, respondents may underreport unhealthy eating patterns or overstate positive behaviors, such as adherence to healthy eating, in an attempt to present themselves in a more favorable light. The use of the CAWI (Computer-Assisted Web Interview) method, while offering advantages like cost efficiency and accessibility, also has implications for sample diversity and reliability. As the study recruited participants online, it inherently relied on internet accessibility and engagement with specific social media platforms or online communities. This approach may have excluded individuals without regular internet access or those less likely to engage in digital environments, potentially leading to a sample skewed toward younger, urban, and more digitally literate individuals. Additionally, the absence of direct interaction between researchers and participants in the CAWI method may have reduced opportunities to clarify questionnaire items, increasing the risk of misinterpretation. While the anonymous nature of online surveys might encourage more honest responses, it also limits the ability to verify the accuracy of self-reported data or address potential ambiguities. Furthermore, the specificity of the studied group—women involved in dietetics and strength training—further limits the generalizability of the findings. The study also did not account for additional contextual factors, such as family pressure, social support, or living arrangements (e.g., living alone, with roommates, or with parents), which could significantly influence eating behaviors and dietary focus.

The study’s innovation lies in the simultaneous analysis of the influence of both dietetics and strength training on ON, which is not widely explored. Most existing research focuses on one of these areas, whereas this study comprehensively combines both elements, analyzing their impact on the risk of ON and body image perception. Furthermore, the inclusion of a specific group—dietetics students and women who engage in strength training—provides new data on potentially at-risk populations that may not be as frequently studied in the context of ON.

The significance of the study stems from its relevance to the growing issue of EDs related to excessive focus on healthy eating and physical activity. The results of this study may have direct implications for professionals involved in public health, dietetics, and mental health, as they highlight potential at-risk groups.

## 5. Conclusions

The study highlights a significant relationship between dietetics involvement, physical activity, and the risk of developing ON. Women associated with dietetics—whether or not they engage in strength training—exhibited a higher ON risk, suggesting that heightened nutritional awareness can lead to pathological eating patterns focused on food purity. This underscores the need for educational interventions that promote balanced dietary behaviors while discouraging obsessive tendencies.

Body image self-esteem plays a protective role against ON, as individuals with higher self-esteem are less prone to excessive dietary control due to greater body acceptance and reduced pressure to achieve an “ideal” body. Conversely, low body image self-esteem, particularly among dietetics students not engaged in strength training, combined with heightened social attribution, increases ON risk by driving efforts to meet external appearance standards.

Interventions targeting body image self-esteem and reducing reliance on social approval may play a role in reducing ON risk, particularly in high-risk groups such as dietetics students and physically active women. Psychological support aimed at promoting realistic self-esteem and body acceptance could be beneficial; however, further research is needed to validate its effectiveness. These strategies could encourage a healthier and more balanced approach to nutrition and physical activity, but their impact should be explored in the context of targeted interventions and specific populations.

## Figures and Tables

**Figure 1 behavsci-15-00199-f001:**
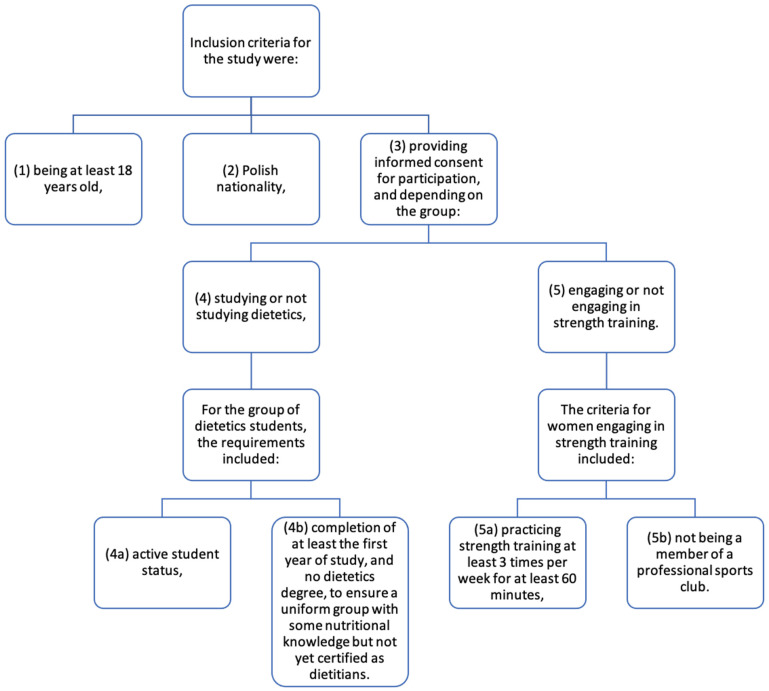
The inclusion and exclusion criteria for the study.

**Figure 2 behavsci-15-00199-f002:**
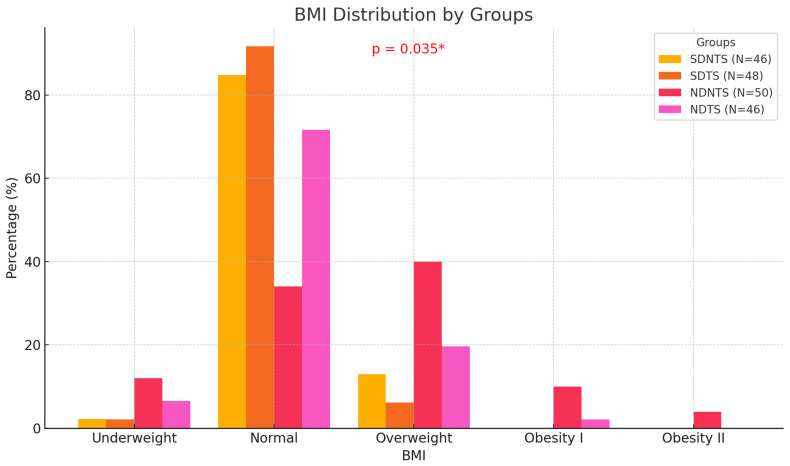
Nutritional status by groups of female respondents according to BMI classification (*N* = 190). BMI—Body Mass Index; SDNTS—dietetics students not engaged in strength training; SDTS—dietetics students engaged in strength training; NDNTS—non-dietetics students not engaged in strength training; NDTS—non-dietetics students engaged in strength training; * = *p* < 0.05.

**Figure 3 behavsci-15-00199-f003:**
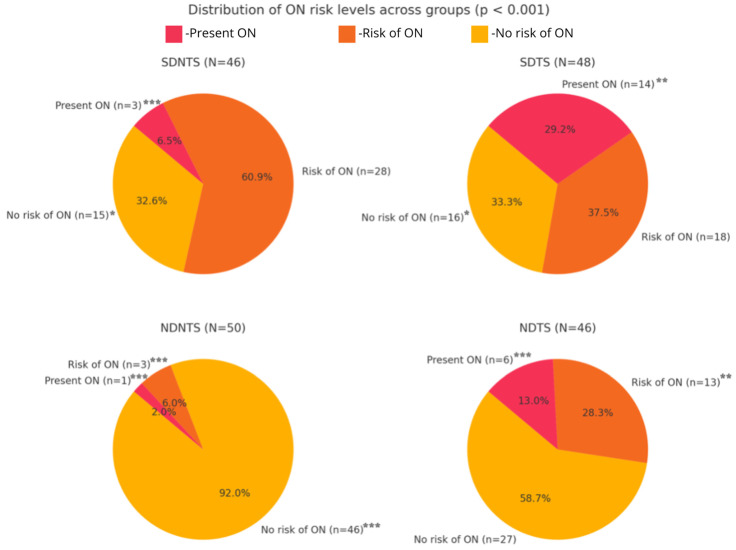
Orthorexia Nervosa (ON) risk interpreted through the DOS scale (*n* = 190) (*p* < 0.001); SDNTS—dietetics students not engaged in strength training; SDTS—dietetics students engaged in strength training; NDNTS—non-dietetics students not engaged in strength training; NDTS—non-dietetics students engaged in strength training; * = *p* < 0.05; ** = *p* < 0.005; *** = *p* < 0.001.

**Figure 4 behavsci-15-00199-f004:**
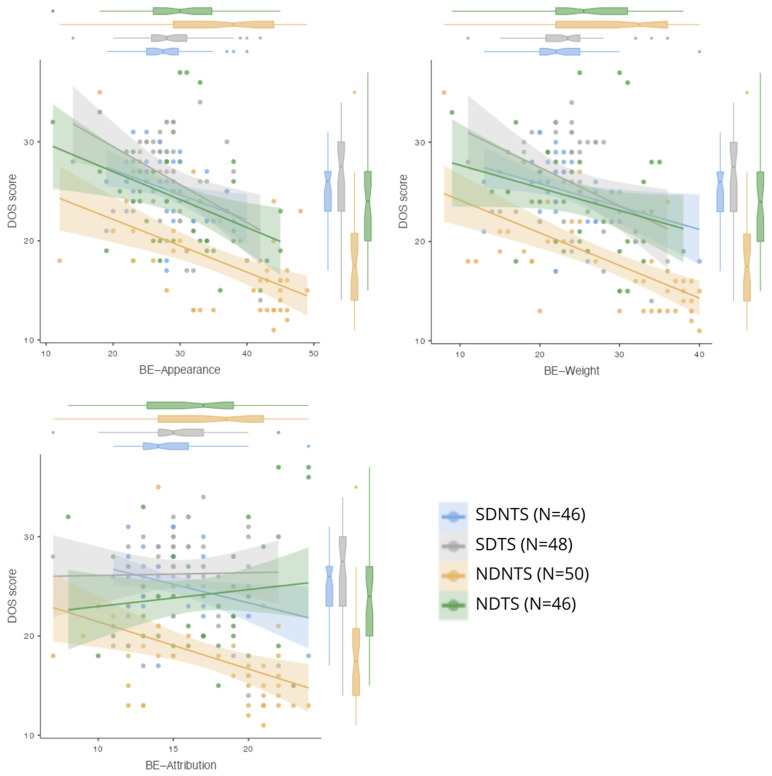
Scatter plots of scores obtained in the BESSA and DOS subscales. BE—body esteem.

**Table 1 behavsci-15-00199-t001:** Characteristics of the study group (*N* = 190).

Variable	Group	X	SD	Min	Max	Med	Skew	Kurt	ε^2^	*p*-Value
Age [Years]	Total (*N* = 190)	23.93	5.80	18	54	23	3.45	12.0	0.00	0.970
	SDNTS (*N* = 46)	22.67	1.44	20	27	23	0.56	1.12		
	SDTS (*N* = 48)	22.75	1.34	20	26	23	−0.12	0.04		
	NDNTS (*N* = 50)	25.52	8.43	18	51	22	1.96	1.51		
	NDTS (*N* = 46)	24.71	7.31	18	54	23	2.93	8.44		
Body mass [kg]	Total (*N* = 190)	63.03	9.82	42	100	62	0.80	1.37	0.04	0.126
	SDNTS (*N* = 46)	61.28	7.12	48	80	60	0.46	0.18		
	SDTS (*N* = 48)	61.12	7.23	50	80	60	0.57	0.08		
	NDNTS (*N* = 50)	66.17	1.27	42	95	67	0.30	−0.08		
	NDTS (*N* = 46)	63.34	1.02	46	100	65	0.35	2.32		
Height [cm]	Total (*N* = 190)	166.02	5.77	150	185	166	0.15	0.17	0.02	0.415
	SDNTS (*N* = 46)	166.21	6.29	150	178	166	−0.30	−0.22		
	SDTS (*N* = 48)	166.52	5.18	158	180	165	0.61	−0.13		
	NDNTS (*N* = 50)	164.62	5.86	153	178	165	0.11	−0.34		
	NDTS (*N* = 46)	166.84	5.64	155	185	167	0.58	1.60		
BMI [kg/m^2^]	Total (*N* = 190)	22.85	3.37	16	37	22	1.13	0.35	0.05	0.031 *
	SDNTS (*N* = 46)	22.23	2.53	18	29	22	1.10	1.15		
	SDTS (*N* = 48)	21.95	1.77	18	27	22	0.33	0.38		
	NDNTS (*N* = 50)	24.42	4.7	16	37	24	0.47	0.15		
	NDTS (*N* = 46)	22.69	3.18	16	33	23	0.65	1.45		

X—mean; SD—standard deviation; Min—minimum; Max—maximum; Med—median; Skew—skewness; Kurt—kurtosis; SDNTS—dietetics students not engaged in strength training; SDTS—dietetics students engaged in strength training; NDNTS—non-dietetics students not engaged in strength training; NDTS—non-dietetics students engaged in strength training; ε^2^—effect sizes; * = *p* < 0.05.

**Table 2 behavsci-15-00199-t002:** Respondents’ answers to the DOS scale questions (*N* = 190).

Response	SDTS *N* (%)	NDTS *N* (%)	NDNTS *N* (%)	SDNTS *N* (%)	V-Cramer	*p*-Value
	(*N* = 48)	(*N* = 46)	(*N* = 50)	(*N* = 46)		
Eating healthy food is more important to me than the pleasure it brings.						
Does not apply to me	12 (25)	18 (39.1)	16 (32)	15 (32.6)	0.29	<0.001 *
Somewhat applies to me	27 (56.2)	20 (43.5)	8 (16)	23 (50)		
Definitely does not apply to me	5 (10.4)	3 (6.5)	25 (50)	4 (8.7)		
Definitely applies to me	4 (8.3)	5 (10.9)	1 (2)	4 (8.7)		
I have certain dietary rules that I follow.						
Does not apply to me	16 (33.3)	20 (43.5)	14 (28)	32 (69.6)	0.34	<0.001 *
Somewhat applies to me	27 (56.2)	21 (45.6)	8 (16)	12 (26.1)		
Definitely does not apply to me	3 (6.2)	0 (0)	8 (16)	1 (2.2)		
Definitely applies to me	2 (4.2)	5 (10.9)	20 (40)	1 (2.2)		
Eating only what is considered healthy gives me pleasure.						
Does not apply to me	5 (10.4)	11 (23.9)	21 (42)	4 (8.7)	0.29	<0.001 *
Somewhat applies to me	27 (56.2)	11 (23.9)	6 (12)	11 (23.9)		
Definitely does not apply to me	16 (33.3)	20 (43.5)	21 (42)	31 (67.4)		
Definitely applies to me	0 (0)	4 (8.7)	2 (4)	0 (0)		
I try to avoid being invited to dinner by friends if I know they don’t pay attention to healthy eating.						
Does not apply to me	9 (18.7)	17 (40)	39 (78)	7 (15.2)	0.34	<0.001 *
Somewhat applies to me	16 (33.3)	16 (34.8)	8 (16)	28 (60.9)		
Definitely does not apply to me	20 (41.7)	9 (19.6)	3 (6)	11 (23.9)		
Definitely applies to me	3 (6.2)	4 (8.7)	0 (0)	0 (0)		
I like that I pay more attention to healthy eating than others.						
Does not apply to me	10 (20.8)	8 (17.4)	4 (8)	7 (15.2)	0.38	<0.001 *
Somewhat applies to me	4 (8.3)	16 (34.8)	22 (44)	5 (10.9)		
Definitely does not apply to me	4 (8.3)	1 (2.2)	20 (40)	1 (2.2)		
Definitely applies to me	30 (62.5)	21 (45.6)	4 (8)	33 (71.4)		
I feel bad after eating something I consider unhealthy.						
Does not apply to me	5 (10.4)	10 (21.7)	22 (44)	8 (17.4)	0.27	<0.001 *
Somewhat applies to me	4 (8.3)	6 (13)	11 (22)	0 (0)		
Definitely does not apply to me	36 (75)	23 (50)	12 (24)	35 (76.1)		
Definitely applies to me	3 (6.2)	7 (15.2)	5 (10)	3 (6.5)		
I feel excluded from social circles due to my strict dietary rules.						
Does not apply to me	17 (35.4)	22 (47.8)	36 (72)	8 (17.4)	0.30	<0.001 *
Somewhat applies to me	19 (39.6)	15 (32.6)	14 (28)	36 (78.3)		
Definitely does not apply to me	9 (18.7)	7 (15.2)	0 (0)	2 (4.3)		
Definitely applies to me	3 (6.2)	2 (4.3)	0 (0)	0 (0)		
My thoughts constantly revolve around healthy eating and influence the organization of my day.						
Does not apply to me	7 (14.6)	13 (28.3)	36 (72)	5 (10.9)	0.40	<0.001 *
Somewhat applies to me	21 (43.7)	14 (30.4)	3 (6)	2 (4.3)		
Definitely does not apply to me	17 (35.4)	17 (37)	9 (18)	39 (84.8)		
Definitely applies to me	3 (6.2)	2 (4.3)	2 (4)	0 (0)		
It is hard for me to resist my dietary rules.						
Does not apply to me	5 (10.4)	8 (17.4)	14 (28)	2 (4.3)	0.24	<0.001 *
Somewhat applies to me	13 (27.1)	16 (34.8)	26 (52)	17 (37)		
Definitely does not apply to me	30 (62.5)	22 (47.8)	8 (16)	27 (58.7)		
Definitely applies to me	0 (0)	0 (0)	2 (4)	0 (0)		
I feel upset after eating unhealthy food.						
Does not apply to me	5 (10.4)	10 (21.7)	12 (24)	1 (2.2)	0.34	<0.001 *
Somewhat applies to me	33 (68.7)	18 (39.1)	3 (6)	32 (69.6)		
Definitely does not apply to me	6 (12.5)	14 (30.4)	30 (60)	6 (13)		
Definitely applies to me	4 (8.3)	4 (8.7)	5 (10)	7 (15.2)		

SDNTS—dietetics students not engaged in strength training; SDTS—dietetics students engaged in strength training; NDNTS—non-dietetics students not engaged in strength training; NDTS—non-dietetics students engaged in strength training; * = *p* < 0.005.

**Table 3 behavsci-15-00199-t003:** Body self-assessment of study participants (*n* = 190).

Variable	Group	X	SD	Min	Max	Med	ε^2^	*p*-Value
BE—Appearance	Total (*N* = 190)	30.7	7.6	11.0	49	29	0.14	<0.001 *
	SDNTS (*N* = 46)	27.7	4.73	19	40	27.55		
	SDTS (*N* = 48)	28.5	5.21	14	42	28.0		
	NDNTS (*N* = 50)	35.9	9.48	12	49	38.0		
	NDTS (*N* = 46)	30.3	7.32	11	45	30.0		
BE—Weight	Total (*N* = 190)	25.1	6.89	8.0	40	24	0.11	<0.001 *
	SDNTS (*N* = 46)	22.6	4.68	13	40	22.0		
	SDTS (*N* = 48)	23.1	4.54	11	36	23.5		
	NDNTS (*N* = 50)	28.9	8.97	8	40	32.5		
	NDTS (*N* = 46)	25.7	6.42	9	38	25.5		
BE—Attribution	Total (*N* = 190)	16.1	3.6	7.0	24	15	0.06	0.006 *
	SDNTS (*N* = 46)	14.9	2.79	11	24	14.0		
	SDTS (*N* = 48)	15.6	3.14	7	22	15.0		
	NDNTS (*N* = 50)	17.3	4.17	7	24	18.5		
	NDTS (*N* = 46)	16.5	3.79	8	24	17.0		

X—mean; SD—standard deviation; Min—minimum; Max—maximum; Med—median; BE—body esteem; SDNTS—dietetics students not engaged in strength training; SDTS—dietetics students engaged in strength training; NDNTS—non-dietetics students not engaged in strength training; NDTS—non-dietetics students engaged in strength training; ε^2^—effect sizes; * = *p* < 0.05.

**Table 4 behavsci-15-00199-t004:** Linear regression analysis of body image attitudes and scores on the DOS scale considering the studied groups.

Model Coefficients—DOS Score; R = 0.693; R^2^ = 0.480					
Predictor	Estimate	SE	t	*p*-Value	Standardized Estimate
Intercept	31.727	1.5465	20.515	<0.001 *	
BE—Appearance	−0.22	0.0853	−2.575	0.011 *	−0.30
BE—Weight	−0.158	0.0906	−1.747	0.082	−0.19
BE—Attribution	0.212	0.1042	2.039	0.043 *	0.14
Group: SDTS vs. SDNTS	1.13	0.8585	1.317	0.190	0.20
Group: NDNTS vs. SDNTS	−5.019	0.9196	−5.458	<0.001 *	−0.89
Group: NDTS vs. SDNTS	−0.427	0.8821	−0.484	0.629	−0.08

SE—standard error; t—t-value; SDNTS—dietetics students not engaged in strength training; SDTS—dietetics students engaged in strength training; NDNTS—non-dietetics students not engaged in strength training; NDTS—non-dietetics students engaged in strength training; * = *p* < 0.05.

**Table 5 behavsci-15-00199-t005:** Linear regression analysis of body image attitudes and scores on the DOS scale considering student status and strength training.

Model Coefficients—DOS Score; R = 0.676; R^2^ = 0.457					
Predictor	Estimate	SE	t	*p*-Value	Standardized Estimate
Intercept	28.409	1.7623	16.12	<0.001 *	
BE—Appearance	−0.266	0.0851	−3.13	0.002 *	−0.36
BE—Weight	−0.135	0.0918	−1.47	0.144	−0.16
BE—Attribution	0.216	0.1061	2.04	0.043 *	0.14
Dietetics:					
Yes-No	3.183	0.6532	4.87	<0.001 *	0.56
Strength Training:					
Yes-No	2.797	0.62269	4.46	<0.001 *	0.49

SE—standard error; t—t-value; * = *p* < 0.05.

## Data Availability

The raw data supporting the conclusions of this article will be made available by the authors upon request.

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
