# Peer review of "The Impact of Nutrition Education, Strength Training, and Body Image Perception on Orthorexia Nervosa Risk: A Cross-Sectional Study in Women"

_behavsci, 2025, doi:10.3390/bs15020199_

Round 1

Reviewer 1 Report

Comments and Suggestions for Authors

The study is very interesting and well-written, but I suggest a few revisions.

Firstly, the authors discuss the overlap between orthorexia nervosa and OCD; however, it would be worthwhile to reference recent developments in the literature showing that this relationship is mediated by food-related concerns (Rossi et al., 2024).

Additionally, in the introduction, the consensus conference (Donini et al., 2022), which outlines the criteria for orthorexia nervosa along with its risk and protective factors, should be mentioned.

The results of the statistical analyses should be improved by including effect sizes (Cohen's d and Cramér's V), which should first be explained in the relevant section (2.3.4).

It is unclear whether the regression coefficients reported in the tables are standardized or not—this should be clarified.

REFERENCES:
Donini, L. M., Barrada, J. R., Barthels, F., Dunn, T. M., Babeau, C., Brytek-Matera, A., … Lombardo, C. (2022). A consensus document on definition and diagnostic criteria for orthorexia nervosa. Eating and Weight Disorders, 27(8), 3695–3711. https://doi.org/10.1007/s40519-022-01512-5

Rossi, A. A., Mannarini, S., Donini, L. M., Castelnuovo, G., Simpson, S., & Pietrabissa, G. (2024). Dieting, obsessive–compulsive thoughts, and orthorexia nervosa: Assessing the mediating role of worries about food through a structural equation model approach. Appetite, 193, https://doi.org/10.1016/j.appet.2023.107164

Author Response

Thank you so much for taking the time to evaluate our work. We have tried to incorporate all your valuable suggestions. If we could improve our work in any way, please let us know.

The study is very interesting and well-written, but I suggest a few revisions..

Comment 1

Firstly, the authors discuss the overlap between orthorexia nervosa and OCD; however, it would be worthwhile to reference recent developments in the literature showing that this relationship is mediated by food-related concerns (Rossi et al., 2024)..

Thank you very much for your suggestion, added.

Comment 2

Additionally, in the introduction, the consensus conference (Donini et al., 2022), which outlines the criteria for orthorexia nervosa along with its risk and protective factors, should be mentioned..

Thank you very much for your suggestion, corrected.

Comment 3

The results of the statistical analyses should be improved by including effect sizes (Cohen's d and Cramér's V), which should first be explained in the relevant section (2.3.4)..

Thank you very much for your suggestion, corrected. ε² and V-Cramer were used.

Comment 4

It is unclear whether the regression coefficients reported in the tables are standardized or not—this should be clarified..

Thank you for your observation. The statistical description has been updated to clarify this point, and the regression coefficients in the tables now include standardized values for better interpretation and comparability.

Thank you very much for your valuable comment, corrected

Thank you for your help. Your guidance is invaluable.

Kind regards,

Authors.

Reviewer 2 Report

Comments and Suggestions for Authors

First of all, I would like to congratulate the authors on the writing of this scientific article.

Starting with the abstract, I find the introduction excessively long, which could allocate more space to describe significant results.
The abstract includes acronyms but does not provide their meanings, which should be corrected:

  • SDNTS – dietetics students who do not engage in strength training
  • SDTS – dietetics students who engage in strength training
  • NDNTS – women who are not studying dietetics and do not engage in strength training
  • NDTS – women who are not studying dietetics but engage in strength training

Regarding the article itself, I also believe that the introduction should be more concise and objective, as it is overly extensive and tires the reader before reaching the methodology section.

Was there a specific reason for selecting only women? If so, what was it?

Was exercise duration a criterion for inclusion?

The methodology section is expected to describe the methods used and not provide justifications. For instance, the second paragraph in section 2.3.3 is not appropriate for the methodology section.

How many participants were initially surveyed, and how many had to be excluded?

Results

The way the results are presented at the end of page 6 and the beginning of page 7 is difficult to follow – a table would be more appropriate.

Table 2 is somewhat confusing, as BMI is based on weight and height. It might be clearer if the table focused solely on results related to this parameter.

Consider revising the layout of Table 3. Perhaps align the text to the left in the first column and bold the phrases, as having phrases embedded among the results makes the table harder to read.

Figure 2 could benefit from a legend explaining the colors of each section in the chart.

The way the BESAA scale is interpreted is not adequately described in the methodology, which makes it difficult to understand the results.

The paragraph on page 20 that starts with "these findings..." and ends with "weight satisfaction alone" belongs in the discussion section, not in the results. The same applies to the paragraph following Table 4.

Discussion

The discussion is well-conceived but should avoid repeating many of the results previously presented (include only the essential ones or rephrase, e.g., instead of stating 25%, say "1 in 4 individuals").

When referencing the studies by Villa et al. and Valente et al., comparisons should be made with your own findings.

Were you unable to find additional studies that used the BESAA scale?

From the fifth paragraph on page 16, the discussion becomes engaging, but it lacks comparisons with other studies.

Conclusion

The conclusion is well-structured; however, the term "ideal physique" should be replaced with "ideal body."

The sentence "Psychological support that fosters realistic self-esteem and body acceptance can help mitigate ON risk, encouraging a healthier and more balanced approach to nutrition and physical activity" should be rewritten. The scales used in the study do not provide sufficient data to support this assertion. Instead, you could suggest psychological interventions with specific groups, but the statement should not be as definitive as it is presented at the end of the paper.

Author Response

Thank you so much for taking the time to evaluate our work. We have tried to incorporate all your valuable suggestions. If we could improve our work in any way, please let us know.

The study is very interesting and well-written, but I suggest a few revisions..

Comment 1

First of all, I would like to congratulate the authors on the writing of this scientific article.

Starting with the abstract, I find the introduction excessively long, which could allocate more space to describe significant results.

The abstract includes acronyms but does not provide their meanings, which should be corrected:

SDNTS – dietetics students who do not engage in strength training

SDTS – dietetics students who engage in strength training

NDNTS – women who are not studying dietetics and do not engage in strength training

NDTS – women who are not studying dietetics but engage in strength training

Thank you very much for your suggestion, corrected.

Comment 2

Regarding the article itself, I also believe that the introduction should be more concise and objective, as it is overly extensive and tires the reader before reaching the methodology section...

The introduction has been shortened, Thank you very much for your suggestion.

Comment 3

Was there a specific reason for selecting only women? If so, what was it?..

The study focused exclusively on women to investigate the unique relationship between body image perception, dietary behaviors, and the risk of ON in this population. Women are often disproportionately affected by body image issues, societal pressures to conform to idealized beauty standards, and eating disorders compared to men. This makes them a particularly relevant group for examining these dynamics. Previous research indicates significant gender differences in the prevalence, manifestation, and predictors of ON. By limiting the study to women, the research aimed to provide a more in-depth and focused analysis of the factors influencing ON risk in this specific demographic..

Comment 4

Was exercise duration a criterion for inclusion?.

Yes, exercise duration was a criterion for inclusion in the study. Participants in the strength-training groups were required to engage in strength training at least 3 times per week for a minimum of 60 minutes per session.

Comment 5

The methodology section is expected to describe the methods used and not provide justifications. For instance, the second paragraph in section 2.3.3 is not appropriate for the methodology section.

.

Thank you very much for your suggestion, corrected.

Comment 6

How many participants were initially surveyed, and how many had to be excluded?

A total of 204 people were included in the group, but 190 participants were included in the final analysis after taking into account the exclusion criteria..

Comment 7

The way the results are presented at the end of page 6 and the beginning of page 7 is difficult to follow – a table would be more appropriate.

Thank you very much for your suggestion, corrected.

Comment 8

Table 2 is somewhat confusing, as BMI is based on weight and height. It might be clearer if the table focused solely on results related to this parameter.?

Thank you for your comment, however, we consider that body mass and height information are often reported in studies and would like to leave them off the table.

Comment 9

Consider revising the layout of Table 3. Perhaps align the text to the left in the first column and bold the phrases, as having phrases embedded among the results makes the table harder to read.

Thank you very much for your suggestion, corrected.

Comment 10

Figure 2 could benefit from a legend explaining the colors of each section in the chart.

Thank you very much for your suggestion, corrected.

Comment 11

The way the BESAA scale is interpreted is not adequately described in the methodology, which makes it difficult to understand the results.

Thank you very much for your suggestion, corrected.

Comment 12

The paragraph on page 20 that starts with "these findings..." and ends with "weight satisfaction alone" belongs in the discussion section, not in the results. The same applies to the paragraph following Table 4..

Thank you for your valuable comment we have corrected the description to be more appropriate for the results section.

Comment 13

The discussion is well-conceived but should avoid repeating many of the results previously presented (include only the essential ones or rephrase, e.g., instead of stating 25%, say "1 in 4 individuals").

Thank you very much for your suggestion, corrected.

Comment 14

When referencing the studies by Villa et al. and Valente et al., comparisons should be made with your own findings.

Added additional description as suggested..

Comment 15

Were you unable to find additional studies that used the BESAA scale?. From the fifth paragraph on page 16, the discussion becomes engaging, but it lacks comparisons with other studies.

Compared as recommended.

Comment 16

The conclusion is well-structured; however, the term "ideal physique" should be replaced with "ideal body."

The sentence "Psychological support that fosters realistic self-esteem and body acceptance can help mitigate ON risk, encouraging a healthier and more balanced approach to nutrition and physical activity" should be rewritten. The scales used in the study do not provide sufficient data to support this assertion. Instead, you could suggest psychological interventions with specific groups, but the statement should not be as definitive as it is presented at the end of the paper..

Thank you for your valuable comments, the conclusion has been corrected.

Thank you for your help. Your guidance is invaluable.

Kind regards,

Authors.